# Norepinephrine Effects on Uropathogenic Strains Virulence

**DOI:** 10.3390/microorganisms10112248

**Published:** 2022-11-14

**Authors:** Nadezhda Ignatova, Alina Abidullina, Olga Streltsova, Vadim Elagin, Vladislav Kamensky

**Affiliations:** 1Department of Epidemiology, Microbiology and Evidence-Based Medicine, Privolzhsky Research Medical University, 603005 Nizhny Novgorod, Russia; 2Department of Urology, Privolzhsky Research Medical University, 603005 Nizhny Novgorod, Russia; 3Institute of Experimental Oncology and Biomedical Technologies, Privolzhsky Research Medical University, 603005 Nizhny Novgorod, Russia

**Keywords:** norepinephrine, *E. coli*, *S. aureus*, biofilm, matrix, adhesion, biomass

## Abstract

The degree of virulence correlates with adhesion, biofilm formation, motility and the capacity to quickly colonize biological surfaces. The virulence of the bacteria that have colonized the urinary tract may be modified by substances dissolved in urine. One such substance is the norepinephrine (NE) hormone, which may be present in human urine, especially in times of stress and under changes in the activity of the renin-angiotensin-aldesterone system. In this work, we study the influence of NE on the biomass, biofilm formation, matrix production, adhesion, motility and metabolism of uropathogenic strains of *E. coli* and *S. aureus*. We used Congo red and gentian violet staining for detection of matrix and biomass formation, respectively. The optical density was measured by a multichannel spectrophotometer. The motility of bacterial cells was measured on semi-solid agar at 24 h and 48 h. The metabolic activity was analyzed by MTT assay. It was shown that the metabolic activity of *E. coli* was stimulated by NE, which led to the increasing synthesis of virulence factors such as biofilm production, adhesion, and motility. At the same time, NE did not activate the *S. aureus* strain’s metabolism and did not change its adhesion and motility. Thus, the virulence activity of uropathogenic *E. coli* may be modified by NE in urine.

## 1. Introduction

Intestine commensals, such as *Escherichia coli* and *Staphylococcus aureus,* are the most common reasons for enteritis and some extra-intestinal diseases, such as urinary tract infections (UTIs) [1,2]. According to statistical data, up to 40% of women and up to 12% of men experienced at least one episode of UTI during their lifetime. In addition to a wide prevalence, UTI is characterized by a tendency to recurrence (from 27% to 48%), which is complicated by the high resistance of strains to available antibacterial drugs [3]. The presence of a certain set of pathogenicity factors helps bacteria to colonize and spread from the initial site of infection. Thus, uropathogenic strains of *E. coli* are characterized by virulence factors such as fimbrial adhesins, cytotoxic necrotic factor type 1 and 2, hemolysins [4,5], high motility, and the capacity for biofilm formation [6,7].

Biofilm formation is one of the most frequent causes of chronic infection, the formation of nosocomial strains, and antibiotic resistance [8]. Biofilms consist of an exopolymeric matrix (up to 85% of the volume) and bacterial cells, which can be represented by mono- or polyculture [9]. The activity of pathogenicity factors may depend on the bacterial cell itself and on its environment, for example, on the hormonal composition of the biofluid in which the bacteria are present. The exacerbation of chronic diseases, a high-calorie diet, a sedentary lifestyle, obesity, hyperinsulinemia, and changes in the activity of the renin-angiotensin-aldesterone system [10] can lead to an increase in the concentration of norepinephrine (NE) in biological fluids. In the human body, it has an impact on almost all systems: it increases the tone of the smooth muscles of the vessels, sphincters of the gastrointestinal tract and urinary tract, and inhibits the secretion of glands [11]. For example, during prolonged stress, an increased level of catecholamines can significantly affect the pathogenicity factors of bacteria [9,12]. We hypothesized that the excretion of hormones in the urine may affect the virulence factors of uropathogenic bacteria. Thus, the goal of the work was to study the effect of catecholamines on the activity of pathogenicity factors of *E. coli* and *S. aureus* strains associated with uroinfections.

## 2. Materials and Methods

### 2.1. Bacterial Strains and Cultivation

The studies were carried out on 12 strains of *E. coli* and 13 strains of *S. aureus* isolated from the surface of urinary concretions and from urine samples of patients with uroinfections undergoing treatment at the Nizhny Novgorod Regional Clinical Hospital named after N.A. Semashko. The species identity was determined by matrix assisted laser desorption ionization-time of flight mass spectrometry (MALDI ToF Autoflex speed, Bruker Daltonik GmbH, Bremen, Germany). Routine cultivation was carried out using nutrient agar (24 h, 37 °C). Nutritious broth was used as a liquid medium. For all experiments, overnight cultures of bacteria on a log phase of growth diluted to 0.5 CFU were used. The final concentration of norepinephrine (NE, Laboratoire Aguettant, Lyon, France) used in the study was 0.052%, which corresponds to the mean physiological level of the catecholamine in the urine of healthy people described previously [13].

### 2.2. Biofilms Formation

The ability of microorganisms to form biofilms was determined by seeding the studied strains on agar containing Congo red [8]. The method is based on the ability of Congo red to stain the polysaccharides, the major component of the biofilm matrix. Strains with a high ability of biofilm formation stain agar around colonies in black, strains with average biofilm formation ability are characterized by staining only colonies in black or brown, and strains that are not capable of biofilm formation do not have colony or medium staining.

### 2.3. Effects on the Growth of Bacterial Biomass

Biofilms were cultured in polystyrene 96-well plates on a nutrient broth with a bacterial concentration of 1 × 10^8^ CFU/mL. To analyze the effect of NE on bacterial biomass growth, it was added to some wells, the control ones were hormone free. Plates with suspension were cultured for 24 h and 48 h at 37 °C. After incubation, the biofilms were washed three times by phosphate buffered saline (PBS), fixed with 96% ethyl alcohol for 15 min, and stained by 0.1% gentian violet solution (3 min). Next, the dye was eluted by 96% ethyl alcohol at constant shaking (10 min) and the optical density was measured using a multichannel spectrophotometer at a wavelength of 570/640 nm.

### 2.4. Matrix Production Assay

To analyze the effect of NE on matrix production, the biofilms were grown for 24 h or 48 h in the presence of hormone. The generated biofilms were washed three times by PBS and stained by Congo red for 15 min. The staining solution contained 1% of Congo red and 10% Twin 80, and were prepared in PBS. The bacterial cells were counterstained by 0.1 gentian violet solution [14]. The biofilms were visualized using a Leica DMIL microscope (Leica, Germany) at a magnification of 1000×.

### 2.5. Bacterial Adhesion Assay

The bacterial cells seeded on polystyrene plates were cultivated during various times (15 min, 30 min, 45 min, 60 min, 3 h or 5 h) in the presence of NE and without it (control). The plates were then rinsed and adhesive cells were stained by gentian violet. To quantify the results, the stain was eluted by 96% ethanol for 10 min and the optical density was measured at 570 nm.

### 2.6. Determination of Bacterial Motility

To study the motility, bacteria were seeded in semi-liquid agar (0.5%) by puncturing the medium. The diameter of the grown colonies was measured in 24 h and 48 h of culturing (37 °C) [6].

### 2.7. MTT Assay

The 3-(4,5-dimethylthiazol-2-yl)-2,5-diphenyltetrazolium bromide (MTT) reduction assay was used for detection of metabolic activity alteration. After 24 h or 48 h the biofilms were washed three times by PBS and MTT solution was added. After 3 h incubation (37 °C), the dye was replaced to 100 µL of dimethyl sulfoxide (DMSO). The optical density of the obtained solutions was measured using a microplate reader at wavelengths of 570 nm and 630 nm.

### 2.8. Statistical Analysis

The data are presented below as the mean values and standard deviations (SD). A statistical analysis was performed with Statistica 10 (StatSoft. Inc., Tusla, OK, USA). The nonparametric Mann-Whitney U-test was used. *p*-values ≤ 0.05 were considered to be statistically significant.

## 3. Results

### 3.1. Determination of the Ability of Uropathogenic Strains to Form Biofilms

Cultivation of microorganisms on Congo red agar showed that 75% (9/12) of *E. coli* strains have an ability to form biofilms, and 25% (3/12) do not produce a polysaccharide matrix and do not stain the Congo agar in black (Figure 1a). Among *S. aureus,* 92% (12/13) of the strains were biofilm-forming (Figure 1b).

### 3.2. NE Effects on Bacterial Biomass

NE affected both bacteria species, but the effect was displayed at different times for *E. coli* and *S. aureus* (Figure 2). 24 h after seeding, the biomass of *E. coli* did not differ in the presence and absence of NE. However, after 48 h of cultivation, the biomass of bacteria grown on the medium with NE was 1.7 times (*p* < 0.05) more than on the medium without NE (Figure 2a). At the same time, the presence of NE in the nutrient medium did not stimulate the growth of *E. coli* strains that do not have the ability to form biofilms. Thus, NE enhanced the growth of the biomass of *E. coli* strains capable of synthesizing a polysaccharide matrix. NE also stimulated the growth of *S. aureus* strains. Maximal bacterial biomass was formed after 24 h of incubation and preserved after 48 h (Figure 2b).

### 3.3. NE Effects on Matrix Production

The matrix production was analyzed for *E. coli* (Figure 3a,b) and *S. aureus* upon incubation with NE for 24 h and 48 h. In 24 h of incubation with NE, significant differences were observed only for *E. coli* strains. We found that the matrix synthesis increased and the architecture of the biofilm in the wells with NE changed (Figure 3). There was an abundance of matrix in the wells with NE in comparison with the control.

In 48 h an increased matrix synthesis was observed in both *E. coli* and *S. aureus* strains (Figure 4).

### 3.4. NE Effects on Metabolic Activity

The metabolic activity was measured in mature biofilms after 24 h and 48 h of growth in the presence of NE in a nutrient medium. It was shown that NE initiated the metabolic activity of *E. coli* strains, especially at 24 h of growth; this effect persisted for 48 h biofilms (Figure 5a,b). NE did not significantly change the metabolic activity of *S. aureus* strains (Figure 5a,b). The depletion of the nutrient medium led to a decrease in the metabolic activity of both species after 48 h.

### 3.5. NE Effects on Adhesion

The ability of uropathogenic strains for adhesion under the treatment with NE was assessed in the time period from 15 min to 5 h.

It was shown that *E. coli* strains had enhanced adhesion in the presence of NE during the whole period of observation with larger differences after 3 h (Figure 6a). In *S. aureus* strains, NE decreased adhesion only for 45 min and did not induce any changes in adhesion during the whole period of observation (Figure 6b).

### 3.6. NE Effects on Motility

The influence of NE on the motility of microorganisms on semi-liquid agar was evaluated by measuring the diameter of the colonies. The diameter of the colonies of *E. coli* strains was 1.2 times larger in the presence of NE after 48 h of culturing (Figure 7a).

The diameter of the colonies of *S. aureus* strains did not change in the presence of NE (Figure 7b). Therefore, the presence of NE in the nutrient medium increased the motility of *E. coli* strains.

## 4. Discussion

UTIs have a wide prevalence, as well as a tendency to transition to recurrent and chronic forms [5]. The ability of bacteria to form biofilms on biotic surfaces plays an important role in these processes. A biofilm consists of a biomass of bacteria and the matrix, which is the basic substance synthesized by them. The process of biofilm formation begins with the primary adhesion of bacteria to the surface and the subsequent formation of multilayer cell clusters due to the production of special protein molecules and polysaccharides [7]. It is known that the degree of adhesion and the ability to produce biofilms correlates with the degree of virulence. In the case of colonization by bacteria of the urinary tract, their virulence can be modified by substances dissolved in urine. One of these substances is NE hormone, which may be present in human urine. In this work, we studied the influence of NE on the biomass growth, biofilm formation, matrix production, adhesion, motility and metabolism of uropathogenic strains of *E. coli* and *S. aureus*.

It was shown that 75% of isolated uropathogenic strains of *E. coli* and 92% of strains of *S. aureus* had the ability to produce biofilms. A high level of biofilm-forming strains among uropathogens is explained by their ability to cause chronic UTIs and urolithiasis. According to the previous studies, up to 80% of uropathogens of various genera are able to produce biofilms [8]. This ability is most often found among gram-negative uropathogens of the *Enterobacteriaceae* family. This can be explained by the fact that bacterial communities use the quorum sensing (QS) system to control biofilm formation. The mechanism works through the autoinductors, which are most actively produced in gram-negative bacteria [15].

The assessment of NE’s effect on the colonization of a polystyrene surface by the bacteria showed that the adhesion of *E. coli* strains was enhanced in contrast to *S. aureus*. A possible explanation is that NE increases intracellular concentration of cyclic-diguanylic acid [16,17] or affects their targets. The absence of the effect in the case of *S. aureus* is probably due to the similarity of NE structure to auto-inductor 3, which is specific for gram-negative species [18]. Due to enhanced adhesion, the biomass of *E. coli* cultured with NE was slightly higher after 24 h. In 24 h the attached bacterial cells of both species were found to intensify metabolic activity in order to increase the production of the extracellular matrix. It is known that when a number of bacterial cells reach a threshold and the signaling molecules (e.g., NE or auto-inductors) then *LasI* and *RhlR* systems promote rhamnolipid formation, required for exopolysaccharide *Psl* and *Pel* formation molecule [19]. Alteration in rhamnolipids production may also regulate the bacterial motility. It was previously shown that the production of rhamnolipids, which are important in swarming motility, decreased on semisolid surfaces upon exposure to NE [19]. In our study, the motility of both species of bacterial cell increased after 48 h of cultivation, when the NE concentration in the nutrient medium is reduced. It should be noted that NE did not have any effect on non-biofilm forming strains. QS bacteria have adrenal-like receptors capable of sensing NE and sensory kinase systems (*QseC*/*QseB*) [9]. The presence of these systems promotes the effects of human stress hormones on bacteria [20]. Therefore, NE in the urine can stimulate QS strains such as *E. coli* or *S. aureus* to form biofilms and persistent infection.

## 5. Conclusions

This study shows that the metabolic activity *E. coli* is stimulated by NE, which led to the increasing synthesis of virulence factors such as biofilm production, adhesion and motility. At the same time, NE did not activate the *S. aureus* strain’s metabolism and did not change itsadhesion or motility. It was also shown that NE does not have an effect on the non-biofilm-forming strains of either *E. coli* or *S. aureus*. Thus, the virulence activity of uropathogenic QS strains of *E. coli* may be modified by NE in urine. NE may stimulate the microorganisms to colonize the urinary tract which leads to persistent infection.

## Figures and Tables

**Figure 1 microorganisms-10-02248-f001:**
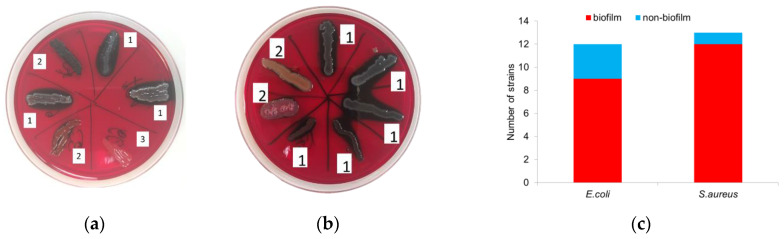
Detection of biofilm-forming uropathogenic strains. Representative images depicting the colonies of *E. coli* (**a**) and *S. aureus* (**b**) in the Congo red agar medium (1–high ability, 2–medium ability, 3–inability to form biofilm) and amount of biofilm-forming strains (**c**).

**Figure 2 microorganisms-10-02248-f002:**
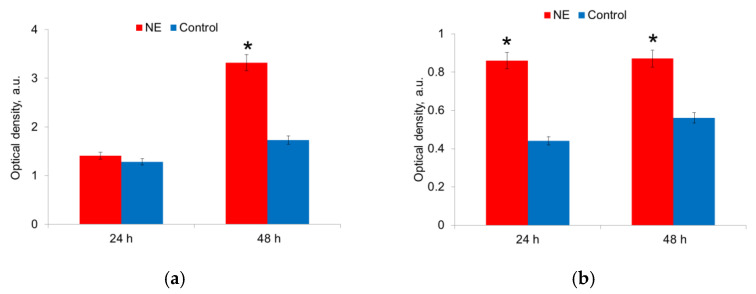
NE effect on biomass of *E. coli* (**a**) and *S. aureus* (**b**) cells in biofilm in 24 h and 48 h. *—statistically significant differences in comparison with the control without NE.

**Figure 3 microorganisms-10-02248-f003:**
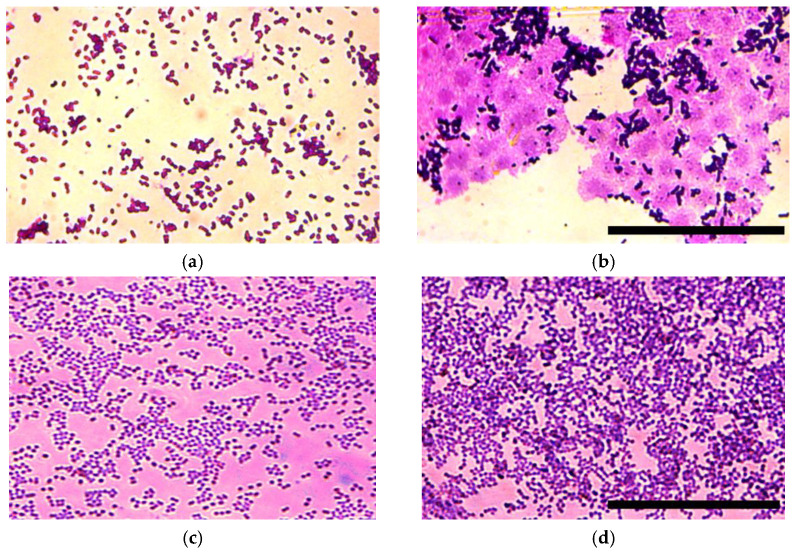
Representative microscopic images depicting the biofilm matrix produced by *E. coli* (**a**,**b**) and *S. aureus* (**c**,**d**) after 24 h of incubation with (**b**,**d**) and without (**a**,**c**) NE. Scale bar is 50 um.

**Figure 4 microorganisms-10-02248-f004:**
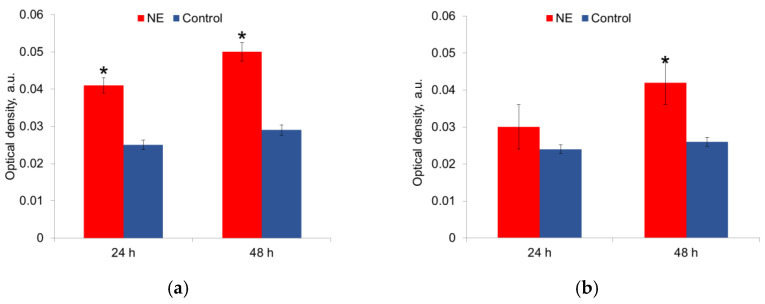
NE effect on matrix production by *E. coli* (**a**) and *S. aureus* (**b**) in 24 h and 48 h of incubation. *—statistically significant differences in comparison with the control.

**Figure 5 microorganisms-10-02248-f005:**
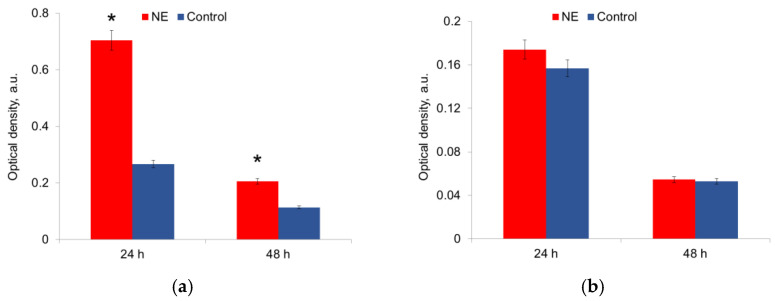
NE effect on metabolic activity of uropathogenic microorganisms *E. coli* (**a**) and *S. aureus* (**b**) in biofilms. The data of MTT-assay are present. *—statistically significant differences in comparison with the control without NE.

**Figure 6 microorganisms-10-02248-f006:**
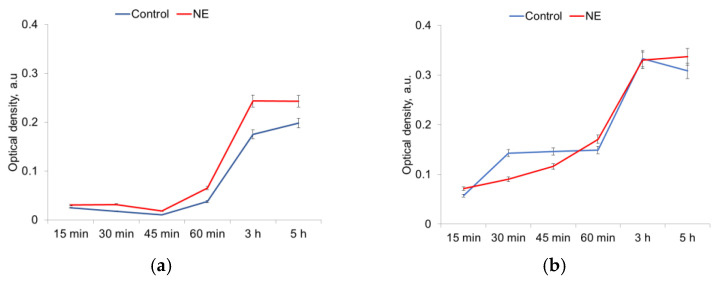
NE effect on adhesion of *E. coli* (**a**) and *S. aureus* (**b**) cells.

**Figure 7 microorganisms-10-02248-f007:**
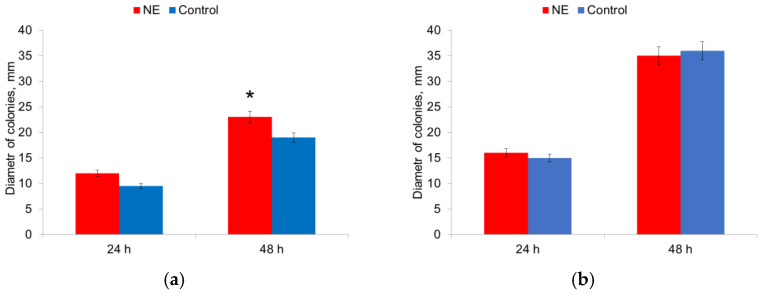
NE effect on motility of *E. coli* (**a**) and *S. aureus* (**b**) cells. *—statistically significant differences in comparison with the control without NE.

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
