# Peer review of "Norepinephrine Effects on Uropathogenic Strains Virulence"

_microorganisms, 2022, doi:10.3390/microorganisms10112248_

Round 1

Reviewer 1 Report

Thanks for the chance to read the article.

I recommend a few modification.

Double check the English by a native English speaker.

Please describe the abbreviations when you use them for the first time.

The paragraph between Matrix production assay is  extremely hard to read, rewrite please.

Figura 1 a is unclear! please send on other size or format!

I  think that you need three short conclusions!

My recommendation is to focus on short conclusions.

Author Response

First of all, we would like to thank editorial board member and reviewers for the very useful comments which allow improving the quality of the manuscript. Enclosed below, please, find our detailed reply to the reviewer comments. Please see the attachment.

Reviewer 2 Report

The manuscript entitled "Norepinephrine effects on uropathogenic strains virulence” reports the influence of Norepinephrine on the biomass, biofilm formation, matrix production, adhesion, mobility and metabolism of uropathogenic strains of E. coli and S. aureus. This is a well conducted study and the text is written concisely and consistently, and the data are sound. The Introduction explained why research was undertaken and the Abstract reflects all relevant aspects of the manuscript. The employed methods are appropriate and were reasonably described. The results can be better presented. The discussion and conclusions are well balanced.

Put all bacterial names in italic in all text.

Page 2 - Material and Methods – Bacterial strains – “…a concentration of 0.5 McFarland…”. Please explain how you determined the concentration of norepinephrine to be used in the culture medium, as well as the characteristics of the healthy person in relation to the drink Page 3 – Figure 1: “… to form biofilm) on Congo agar…”

In figure 1a, point out which samples from the Petri dish are E. coli and S. aureus.

Figure 2 - Does the data represent the average of the 12 strains of E. coli and the 13 strains of S. aureus in all figures? Please explain in the text.

Figure 3 – Please post an image of the biofilm matrix produced by S. aureus or make a comment. Comment if all strains of the same species had the same behavior.

Please confirm the optical density values pointed on the Y axis of figures 4 a and b. They are very reduced.

“NE effects on metabolic activity” - please correct: “…(Fig. 5a). NE did not…(Fig. 5b)…”

“NE effects on adhesion” – “…After 3h of incubation we found statistically significant increasing (p ≤ 0.05)...”

Discussion: Please explain further the role of norepinephrine in adhesion and biofilm formation.

Rewrite the last paragraph of the discussion. It's not very clear.

Author Response

(The authors gave the same response as above.)
